# Marine-Derived Natural Product HDYL-GQQ-495 Targets P62 to Inhibit Autophagy

**DOI:** 10.3390/md21020068

**Published:** 2023-01-20

**Authors:** Quanfu Li, Jianjun Fan, Yinghan Chen, Yiyang Liu, Hang Liu, Wei Jiang, Dehai Li, Yongjun Dang

**Affiliations:** 1Key Laboratory of Metabolism and Molecular Medicine, The Ministry of Education, Department of Biochemistry and Molecular Biology, School of Basic Medical Sciences, Shanghai Medical College, Fudan University, Shanghai 200032, China; 2Department of Immunology, School of Basic Medical Sciences, Shanghai Medical College, Fudan University, Shanghai 200032, China; 3Center for Novel Target and Therapeutic Intervention, Institute of Life Sciences, Chongqing Medical University, Chongqing 400016, China; 4Key Laboratory of Marine Drugs, Chinese Ministry of Education, School of Medicine and Pharmacy, Ocean University of China, Qingdao 266003, China

**Keywords:** EGFP^KI^-LC3B, marine-derived compound, HDYL-GQQ-495, P62, autophagy

## Abstract

Autophagy is widely implicated in pathophysiological processes such as tumors and metabolic and neurodegenerative disorders, making it an attractive target for drug discovery. Several chemical screening approaches have been developed to uncover autophagy-modulating compounds. However, the modulation capacity of marine compounds with significant pharmacological activities is largely unknown. We constructed an EGFP^KI^-LC3B cell line using the CRISPR/Cas9 knock-in strategy in which green fluorescence indicated endogenous autophagy regulation. Using this cell line, we screened a compound library of approximately 500 marine natural products and analogues to investigate molecules that altered the EGFP fluorescence. We identified eight potential candidates that enhanced EGFP fluorescence, and HDYL-GQQ-495 was the leading one. Further validation with immunoblotting demonstrated that cleaved LC3 was increased in dose- and time-dependent manners, and the autophagy adaptor P62 showed oligomerization after HDYL-GQQ-495 treatment. We also demonstrated that HDYL-GQQ-495 treatment caused autophagy substrate aggregation, which indicated that HDYL-GQQ-495 serves as an autophagy inhibitor. Furthermore, HDYL-GQQ-495 induced Gasdermin E (GSDME) cleavage and promoted pyroptosis. Moreover, HDYL-GQQ-495 directly combined with P62 to induce P62 polymerization. In P62 knockout cells, the cleavage of LC3 or GSDME was blocked after HDYL-GQQ-495 treatment. The EGFP^KI^-LC3B cell line was an effective tool for autophagy modulator screening. Using this tool, we found a novel marine-derived compound, HDYL-GQQ-495, targeting P62 to inhibit autophagy and promote pyroptosis.

## 1. Introduction

Marine biological resources have become an important source of lead compounds in new drug research and development because of their diversity, complexity, and particularity. Natural products have a wide range of biological activities and possess key functions across many types of diseases, including cancer, AIDS, Alzheimer’s disease, and malaria. However, there are several steps that stand in the way of natural products entering the clinical stage, such as the ambiguity of molecular targets and the lack of research on their mechanisms of action. Moreover, it is necessary to establish various discovery systems of active natural products to promote natural product-based drug discovery and development.

Autophagy dysfunction is related to the pathogenesis of many human diseases [1,2]. Therefore, autophagy has served as the “hot” target for drug and active natural compound screening. Currently, many in vitro and in vivo screening strategies have been applied to autophagy. The variation in autophagy markers can be detected by fluorescence, Western blotting, flow cytometry, and multifunctional microplate reader. Among these studies, exogenous GFP-LC3 protein was the most used marker [3,4,5]. Based on image acquisition and analysis using high-connotation instruments, the variation of autophagy particle numbers by quantifying the GFP points within each cell or mean fluorescence intensity was measured. Although GFP-LC3 is a simple and widely used screening marker, it cannot distinguish between autophagosomes and autophagy lysosomes. As autophagy is a dynamic and multi-step process, it is necessary to monitor autophagy flux in order to fully evaluate the state of autophagy. Therefore, positive results from primary GFP-LC3 screening undergo rigorous secondary analysis, including analysis of the formation and maturation of autophagosomes and the clearance of autophagy substrates.

To overcome the deficiency of GFP-LC3, researchers have used tandem fluorescence-labeled mRFP-GFP-LC3 to distinguish autophagosomes from autophagy lysosomes. When mRFP-GFP-LC3 is overexpressed in cells, autophagosomes display both mRFP and GFP signals. However, autolysosomes only emit mRFP signals because the mRFP-GFP-LC3 is sensitive to pH, and the GFP signals are easily quenched in acidic environments of autolysosomes [6]. Pampaloni screened the natural compound library and identified six effective autophagy inducers and four inhibitors using the mRFP-GFP-LC3 marker [7]. The mRFP-GFP-LC3 screening requires proper acidification of lysosomes, and the lysosomes may be affected by lysosomal nutrients. Therefore, autophagy substrate clearance and other secondary tests are needed.

A new autophagy probe, GFP-LC3-RFP-LC3△G, was used to evaluate autophagy flux and high-throughput screening [8]. When Atg4 proteases are expressed in cells, the probe can be cleaved into equimolar amounts of GFP-LC3 and RFP-LC3△G. The GFP-LC3 of autolysosomes is degraded, while RFP-LC3△G cannot be lipidated because of its deficiency in C-terminal glycine. Therefore, it can be used as a control in the cytoplasm. The fluorescence intensity of the GFP/RFP ratio was used to measure the autophagy flux. The GFP-LC3-RFP-LC3△G probe has been shown to be useful in measurement of autophagy flux in zebrafish and transgenic mice [8], which is valuable for monitoring autophagy flux in vivo. However, the two LC3 sequences of GFP-LC3-RFP-LC3△G can be homologously recombined to produce GFP-LC3△G that is resistant to degradation by autophagy in retrovirus-infected cells, impacting the screening results.

Elaiophylin, an inhibitor of autophagy, was screened by using GFP-LC3B. Elaiophylin could disrupt the maturation of cathepsin B/D and promote lysosomal membrane permeabilization to inhibit autophagy [4]. Fluspirilene, trifluoperazine, pimozide, niguldipine, nicardipine, amiodarone, loperamide, and penitrem A were also screened using LC3-GFP as a readout. These compounds could promote autophagic degradation. However, it is unclear how these compounds induced autophagy [5]. Nonactin, a new and potent autophagy-inducer, was screened by using an mRFP-GFP-LC3 protein probe. Nonactin is a modulator of energy metabolism in cells, but the specific mechanism of its influence on autophagy remains unknown [7]. Almost all established cell lines and transgenic animals within ectopic expression of GFP-LC3 or mRFP-GFP-LC3 reporter were constitutively expressed under strong promoters such as CMV and CAG promoters. These strategies ignore the effects of LC3 overexpression on autophagy itself and autophagy-related biological processes. Therefore, it is still controversial whether the overexpressed GFP-LC3 or RFP-LC3 in vitro can accurately indicate the autophagy activity of cells. A more ideal method of constructing the LC3 reporter gene is to knock-in the GFP or RFP gene into the coding frame upstream of the endogenous LC3, allowing the expression of the GFP-LC3 or RFP-LC3 fusion proteins driven by the endogenous promoter without altering the expression of the original gene [9].

We constructed an EGFP^KI^-LC3B cell line using the CRISPR/Cas9 system to screen a marine-derived compound library to investigate autophagy modulators. We found that HDYL-GQQ-495, which is chemically modified from the marine compound aspergiolide A [10], serves as a novel autophagy inhibitor. Interestingly, HDYL-GQQ-495 directly combines with P62 to induce P62 polymerization to inhibit autophagy. P62, an autophagy adaptor, is involved in the proteasomal degradation of ubiquitinated proteins. It can shuttle between the nucleus and cytoplasm to bind with ubiquitinated cargoes and facilitate nuclear and cytosolic protein quality control [11]. The oligomerization of P62 via the Phox1 and Bem1p (PB1) domain is important for ubiquitinated protein accumulation [12]. Polymerization of P62, depending on its PB1 domain, is required for its aggregation, resulting in dysfunctional mitochondria and polyubiquitinated proteins degraded by autophagy [13]. It has been reported that verteporfin may act directly on the P62 oligomerization by low singlet oxygen production and inhibits autophagosome accumulation [14,15].

Autophagy blockage promotes pyroptosis by modulating the P62/Nrf2/ARE axis [16]. Pyroptosis, a type of cell death that has been recently elucidated, is an inflammatory programmed cell death regulated by the Gasdermin (GSDM) family and characterized by the formation of plasma membrane pores and the release of intracellular contents [17]. GSDM family proteins contain GSDMA, GSDMB, GSDMC, GSDMD, GSDME, and Pejvakin, which can form membrane pores and destroy the integrity of the cell membrane [18]. GSDME plays an important role in the induction of pyroptosis by chemotherapeutic drugs, cold atmospheric plasma, photodynamic therapy, and CAR T cells [19,20,21,22]. It has been reported that cysteinyl aspartate specific proteinase-3 (caspase-3) and the granzyme-dependent non-classical inflammation mediated by GSDME pathway triggers pyroptosis in tumors [23,24]. Caspase-3 activation was promoted by inhibition of autophagy, despite the observed reduction in cell death, suggesting that autophagy suppresses apoptosis simultaneously activating a caspase-independent cell death pathway [25,26]. Moreover, we found that HDYL-GQQ-495 facilitates pyroptosis through the cleavage of caspase-3 and GSDME in a P62-dependent manner. In summary, we used a CRISPR/Cas9 knock-in strategy to construct the endogenous autophagy reporter cell lines to screen the marine-derived compound library and uncovered that HDYL-GQQ-495 targets P62 polymerization to inhibit autophagy and promote pyroptosis.

## 2. Results

### 2.1. EGFP-LC3B Knock-In Cell Line Construction for Screening of Marine-Derived Compounds

To develop an autophagy modulator, we established EGFP^KI^-LC3B cells using a CRISPR/Cas9-mediated homology-independent targeted integration (HITI) knock-in strategy (Figure 1A). Seven single-cell clones were named 293T EGFP^KI^-LC3B 1 to 7, and the EGFP insertion was confirmed by genotyping and selected for further validation and characterization (Appendix A). The expression of endogenous EGFP-LC3B protein was regulated by its own promoter. Green fluorescence therefore reflected endogenous expression levels and the location of LC3B protein. Rapamycin and chloroquine (CQ) are positive autophagy modulators that lead to the accumulation of autophagosomes and the eventual increase in green fluorescence and EGFP-LC3. As shown in Figure 1B and Appendix A, most of the green fluorescence was distributed throughout the cytoplasm of 293T EGFP^KI^-LC3B cells. After treatment with rapamycin or CQ, the EGFP-LC3 fluorescence intensity and level of autophagy particles in 293T EGFP^KI^-LC3B cells were both significantly increased compared with the control (*p* < 0.001) (Appendix A). Due to the inhibition of autophagy by CQ, EGFP-LC3B was detected by anti-LC3 or anti-GFP antibodies (Figure 1C) in the indicated cell lines. Western blotting analysis revealed the appearance of bands corresponding to an EGFP-LC3B fusion protein, indicating the existence of the functional EGFP-LC3 protein. These results confirmed the correct integration of EGFP at the LC3B locus and the subsequent generation of functional EGFP-LC3B fusion protein.

The 293T EGFP^KI^-LC3B cells were then used to screen compounds that may modulate autophagy. Fluorescence punctate signals representing the accumulated EGFP-LC3 labeled autophagosomes within this region were identified and analyzed. The ability of candidate compounds to modulate autophagy was based upon the average number of observed autophagosome puncta and the average fluorescence intensity in each cell. About 500 marine-derived compounds were detected initially, and the compounds with relative mean GFP fluorescence higher than 120 were thought to be the positive hits (Appendix A). These compounds were then further verified based upon the resultant increase in the intensity of EGFP. The compound HDYL-GQQ-495 was the leading one among the 500 compounds (Figure 1E and Appendix A). Autophagy modulation was further confirmed by the cleaved LC3 protein. As shown in Figure 1F, the ratio of LC3-II/LC3-I was significantly increased (*p* < 0.01) after treatment with the indicated candidate compounds. Interestingly, the autophagy adaptor P62 showed clear bands at the top of P62 monomer only after HDYL-GQQ-495 treatment. Upon investigation of the cytotoxicity of HDYL-GQQ-495 on 293T EGFP^KI^-LC3B cells, we determined that the half-maximal inhibitory concentration (IC_50_) of HDYL-GQQ-495 was 2.30 ± 0.17 µM (Figure 1G). 

In summary, we constructed 293T EGFP^KI^-LC3B cells to screen a library of marine-derived compounds. HDYL-GQQ-495 was the top candidate and could enhance P62 accumulation. Thus, we focused on HDYL-GQQ-495 in further investigation.

### 2.2. HDYL-GQQ-495 Is a Novel Autophagy Inhibitor

HDYL-GQQ-495 is an anthraquinone derivative that is chemically modified from the marine compound aspergiolide A (Figure 2A) [10]. The procedure for the synthesis of HDYL-GQQ-495 is shown in Appendix A. Interestingly, HDYL-GQQ-495 is a fluorescent reagent with an excitation maximum at 500 nm and emission maximum at 590 nm, allowing it to display both red and green fluorescence (Appendix A). The 293T EGFP^KI^-LC3B cells were treated with HDYL-GQQ-495 for 24 h, and the autophagy particles within the treated cells were significantly increased (*p* < 0.01) (Appendix A). Furthermore, we found that LC3II and P62 protein levels increased in a dose- and time-dependent manner with HDYL-GQQ-495 treatment. Interestingly, we uncovered polymerization bands at the top of the P62 monomer, which also increased in intensity in response to increased dose or exposure time (Figure 2B,C). To further verify the effect of HDYL-GQQ-495 on autophagy, we treated HeLa cells with HDYL-GQQ-495 for 24 h and found that the autophagy substrate NBR1 level significantly increased (*p* < 0.001) (Figure 2D and Appendix A). As P62 is well recognized as a selective autophagy receptor and a ubiquitin sensor, it can co-aggregate with ubiquitinated substrates. Therefore, we examined ubiquitinated proteins after HDYL-GQQ-495 treatment and found that ubiquitinated protein levels were dramatically increased (*p* < 0.05) (Figure 2E and Appendix A). The accumulation of P62 protein after the treatment with HDYL-GQQ-495 was further verified by endogenous P62 immunostaining (Figure 2F). All of these results indicate that HDYL-GQQ-495 is a novel autophagy inhibitor, capable of inducing P62 polymerization.

### 2.3. HDYL-GQQ-495 Inhibits Autophagy and Promotes Pyroptosis

Pyroptosis is a type of cell death that has been recently elucidated. It has been reported that autophagy blockage promotes pyroptosis [16], and therefore, we investigated the effect of HDYL-GQQ-495 on pyroptosis. HeLa cells were treated with different concentrations of HDYL-GQQ-495 and incubated for 24 h. Large bubbles from the plasma membrane and cell swelling, morphological features of pyroptosis, were present after this incubation (Figure 3A). Furthermore, the onset of pyroptosis occurred rapidly (6 h and 12 h) after the HDYL-GQQ-495 treatment (Figure 3B). Caspase-3 mediates the cleavage of GSDME to promote pyroptosis in tumors [23,24]. The levels of GSDME and caspase 3 were induced in cells treated with HDYL-GQQ-495 (Figure 3C). Taken together, these data suggest that HDYL-GQQ-495 is a novel autophagy inhibitor and promotes pyroptosis.

### 2.4. HDYL-GQQ-495 Targets P62 to Inhibit Autophagy and Promote Pyroptosis 

P62/SQSTM1 is a multifunctional scaffolding protein involved in the regulation of various signaling pathways as well as autophagy [1]. Its multiple structural domains allow it to be connected to both autophagy and the ubiquitin proteasome system. P62 was observed to be increased and polymerized after HDYL-GQQ-495 treatment, which prompted us to make the hypothesis that P62 may be the ultimate target of HDYL-GQQ-495. We treated purified P62 protein with 10 μM, 20 μM, and 40 μM HDYL-GQQ-495 and found that the polymerized P62 was increased in a dose-dependent manner, indicating that HDYL-GQQ-495 could induce polymerization of purified P62 protein (Figure 4A). Subsequent binding affinity measurement by microscale thermophoresis (MST) showed the direct interaction between MBP-P62 and HDYL-GQQ-495 with a K_d_ of 11.18 ± 4.07 μM. MBP was the negative control (Figure 4B,C). We treated the 293T cell lysate with different concentrations of HDYL-GQQ-495 and found that it also induced the oligomerization of P62 protein within the cell lysate (Figure 4D). These data further confirm our assumption that HDYL-GQQ-495 directly targets P62 to induce its polymerization.

To verify the effect of P62 on autophagy and pyroptosis, we used wild-type and P62 knockout MEF cells. We treated wild-type and P62 knockout MEF cells with the indicated concentrations of HDYL-GQQ-495. In wild-type cells, a remarkable accumulation of LC3II and obvious oligomerization of P62 protein were observed in the HDYL-GQQ-495-treated group compared with the control group. However, there was no accumulation of LC3II protein in P62 KO MEF cells (Figure 4E). In addition, we found that P62 knockout MEF cells exhibited a higher tolerance to HDYL-GQQ-495 treatment than wild-type cells (Figure 4F). Furthermore, the cleavage of GSDME was induced in a dose-dependent manner in the wild-type cells treated with HDYL-GQQ-495, whereas P62 knockout MEF cells did not exhibit this same dose dependency (Figure 4G). These data reinforce the assertion that HDYL-GQQ-495 inhibits autophagy and promotes pyroptosis by targeting P62.

## 3. Discussion

In this study, we constructed a knock-in EGFP-LC3B reporter gene in 293T cells to accurately reflect endogenous autophagy activity using the CRISPR/Cas9 system (Figure 1). Using this cell line, we screened the marine-derived compound library and obtained HDYL-GQQ-495, which could induce P62 polymerization to inhibit autophagy and promote pyroptosis (Figure 2, Figure 3 and Figure 4). This study not only offers a new screening strategy for targeting autophagy but also proves that HDYL-GQQ-495 is a new autophagy inhibitor with new molecular skeletons and a precise target, P62.

Autophagy plays opposing, context-dependent roles in inhibiting tumor initiation and supporting tumor progression. Interventions to modulate autophagy have therefore been proposed as cancer therapies [27]. Numerous discoveries have been reported related to chemical interventions targeting autophagy in cancer. Moreover, several marine-derived compounds with new molecular skeletons have been reported to modulate autophagy in mammalian cells, such as Chromomycin A2, Psammaplin A, ilimaquinone, Jaspine B, and Frondoside A [28,29,30]. However, new molecules and precise targets in the autophagy process are still required to advance clinical drug discovery. 

Benefitting from the CRISPR/Cas9 system, we added the EGFP gene sequence into the first exon of the N terminus of LC3B in 293T EGFP^KI^-LC3B, allowing the endogenous LC3B promoter to drive expression (Figure 1A). The endogenous EGFP-LC3B reporter gene is able to accurately reflect autophagy activity compared with the overexpressed EGFP-LC3B reporter [8]. The 293T EGFP^KI^-LC3B cells had very weak basal levels of green fluorescence without any stimulation. However, the green fluorescence was significantly enhanced in 293T EGFP^KI^-LC3B cells when stimulated with an autophagy activator or inhibitor, especially after treatment with the autophagy inhibitor chloroquine (Figure 1B). This further validates that the constructed cell line can truly reflect the autophagy status of cells. Moreover, the signal-to-noise ratio of 293T EGFP^KI^-LC3B cells is relatively low, which is favorable for screening.

Autophagic flux is important for measuring autophagy activity and studying the role of autophagy. The mRFP-GFP-LC3 tandem probe is a simple, quantitative, and effective method for evaluating the autophagic flux of cells. Autophagic flux can be evaluated by calculating the ratio of GFP/RFP signal values [31]. HDYL-GQQ-495 displays both red and green fluorescence in cells upon excitation at 500 nm with emission readout at 590 nm (Appendix A), leading to the ineffectiveness of the mRFP-GFP-LC3 tandem probe for monitoring autophagy flux. We found that P62 accumulated and polymerized by Western blotting extracts from cells treated with HDYL-GQQ-495 (Figure 2B,C,F). It has been reported that autophagy inhibition increases P62 accumulation, and excess P62 inhibits the clearance of ubiquitinated proteins [32]. Moreover, P62 and ubiquitinated proteins dramatically accumulated after HDYL-GQQ-495 treatment (Figure 2E), which indicated that HDYL-GQQ-495 is an autophagy inhibitor.

When HDYL-GQQ-495 was used to treat cells, cell lysate, and P62 protein, it was found that with increasing concentrations of HDYL-GQQ-495, P62 protein polymerized, and ubiquitinated proteins accumulated (Figure 2A–F and Figure 4A,D). MST assay results showed that HDYL-GQQ-495 directly binds to MBP-tagged P62, compared with the MBP control (Figure 4B,C). P62 interacts with diverse signaling proteins to regulate a variety of cellular functions. Mounting evidence supports an important role of P62 in tumorigenesis, and P62 may become a therapeutic antitumor target [33]. However, compounds targeting P62 activity are very scarce. P62XIE3, XRK3F2, and XIELP1-17b, which share a similar structure, are inhibitors targeting the ZZ-type zinc finger domain of P62. These proteins exhibit antitumor activity in human multiple myeloma cell lines [33,34]. K67, an inhibitor of Keap1–phosphorylated-P62 interaction [35], is structurally analogous to Cpd16, which has previously been identified as a non-covalent inhibitor of the Keap1–Nrf2 interaction [36]. Selective inhibitors of the Keap1–phosphorylated-P62 interaction may function as therapeutic agents for hepatocellular carcinoma (HCC) [37]. HDYL-GQQ-495 is a novel P62 modulator, and no marine compounds targeting P62 have been reported thus far. Further study of the mechanism of action of HDYL-GQQ-495 induction of P62 polymerization will further our understanding of P62-dependent dynamic processes within autophagy.

We found that HDYL-GQQ-495 is capable of leading to autophagy inhibition, caspase-3 activation, and cleavage of GSDME. Caspase-3 is activated after autophagy inhibition [25,26]. However, the mechanism of this activation and inhibition is still unknown. Nevertheless, the cleavage of GSDME requires caspase-3 activation. When GSDME is highly expressed in tumor cells, active caspase-3 cleaves GSDME, and the N-terminal domain lodges itself within the cell membrane to punch holes, resulting in cell swelling, rupture, and death [38]. Blocking of autophagy promotes the pyroptosis of macrophages overloaded with oxidized low-density lipoprotein (ox-LDL) via the P62/Nrf2/ARE axis, which may inspire a new strategy for the clinical application of drugs that inhibit autophagy in atherosclerosis improvement [16]. The combination of ionizing radiation and chemotherapy or demethylated agents induces pyroptosis in tumor cells by promoting CD8^+^ T-cell tumor infiltration and activation of the antitumor response [39]. The natural product triptolide eliminates head and neck cancer cells by inducing GSDME-mediated pyroptosis; however, the mechanism of this remains unclear [40]. We found a new anthraquinone derivative, HDYL-GQQ-495, which may promote pyroptosis in HeLa cells by targeting P62 and inhibiting autophagy. Further investigation of HDYL-GQQ-495 will be valuable for tumor immunotherapy.

## 4. Materials and Methods

### 4.1. Antibodies and Reagents 

The following antibodies were used for Western blotting (WB): GAPDH (clone 1E6D9, 60004-1-Ig, mouse, Proteintech, Rosemont, USA; 1:10,000 for WB), P62/SQSTM1 (18420-1-AP, Rabbit, Proteintech; 1:5000 for WB), β-Tubulin (10094-1-AP, Rabbit, Proteintech; 1:2000 for WB), GFP (50430-2-AP, Rabbit, Proteintech; 1:2000 for WB), LC3B (clone D11, 3868S, Rabbit, Proteintech; 1:1000 for WB), Brilliant Violet 421™ AffiniPure Donkey Anti-Rabbit IgG (H+L) (711-675-152, Jackson ImmunoResearch, PA, USA; 1:10,000 for WB), Donkey anti-mouse IgG (H&L) (715-035-150, Jackson ImmunoResearch; 1:10,000 for WB), and Goat Anti-Rabbit IgG (H+L) (111-035-003, Jackson ImmunoResearch; 1:10,000 for WB). 

HDYL-GQQ-495 was from the Prof. Dehai Li Lab, and the available information of 500 molecules is shown in Appendix A. The following reagents were used: Chloroquine (CQ) (Sigma, MI, USA, cat. no. C6628), Rapamycin (Selleck, Texas, USA, cat. no. S1039), fetal bovine serum (Thermo Fisher, MA, USA, cat. no. 16140071), Penicillin/streptomycin (Thermo Fisher, cat. no. 15140122), RPMI 1640 (Thermo Fisher, cat. no. 11875093), DMEM (Thermo Fisher, cat. no. 11995065), Puromycin Dihydrochloride (Sigma, cat. no. P8833), Trypsin 0.25% EDTA (Thermo Fisher, cat. no. 25200072), Polyethylenimine Hydrochloride (MW 40,000) (Polysciences, PA, USA, cat. no. 24765-1), P62;SQSTM1 Fusion Protein (Proteintech, cat. no. Ag13131), HiScript II 1st Strand cDNA Synthesis Kit (+gDNA wiper) (Vazyme, Nanjing, China, cat. no. R212-01/02), and ChamQ SYBR qPCR Master Mix (Vazyme, cat. no. Q311-02).

### 4.2. Cell Culture

Human HEK293T (ATCC No. CRL-3216^TM^) and HeLa (ATCC No. CRM-CCL-2^TM^) cells were obtained from the American Type Culture Collection (Manassas, VA, USA). HEK293T or HeLa cells were cultured in DMEM (Thermo Fisher) supplemented with 10% fetal bovine serum (FBS) and 1% penicillin/streptomycin/glutamine (Thermo Fisher) in a humidified 5% CO_2_ incubator at 37 °C. The MEF P62 knockout cell line was obtained from the Ronggui Hu Lab. Mammalian cell transfection was performed using Polyethylenimine Hydrochloride (MW 40,000) Transfection Reagent (Polysciences) following the manufacturer’s protocol. 

### 4.3. CRISPR/Cas9 and EGFP-Donor Construction

A 100–200 bp region in which the Cas9 target site should be located was identified. A 200 bp window centered on the start codon (for N-terminal tags) or stop codon (for C-terminal tags) was generally used. This genomic sequence was submitted to the Zhang lab’s CRISPR design tool at http://crispr.mit.edu. A list of potential targeting sequences was returned by the design tool, ranked in order of predicted specificity. Sites with a specificity score >95, and in most cases a site that scores 98 or 99 (100 indicates perfect specificity), were selected. If there were several candidate sites with high specificity, the site that was closest to the desired insertion site (start or stop codon) was selected. 

To the GFP N terminus and C terminus were added 23 nucleotides, which was the gRNA’s reverse complement sequence. Primers were designed to amplify the fragment (the gRNA reverse complement sequence + the interested fluorescent protein + gRNA’s reverse complement sequence). Using Hind III and EcoRI, 4 µg of the pUC19 plasmid was digested. A digested pLentiCRISPR was gel purified using QIAquick Gel Extraction Kit. Following the Vazyme Biotech Co., Ltd. protocol (ClonExpress II One Step Cloning Kit), a new donor was generated. For more information, please refer to [9]. The gRNA, donor, and genotyping target sequences are shown in Table 1.

### 4.4. Transfection and Single Clone Isolation 

The 293T cells were co-transfected with lentiCRISPR v2-sgRNA-LC3B and EGFP-Donor using the PEI reagent for 3 days. The transfected cells were isolated into fluorescent protein + populations by FACS via BD Influx (BD), approximately 200 cells were plated onto 100 mm dishes, and approximately 500 cells were plated onto 150 mm dishes. After 2 weeks, genome-edited HEK293 clones were manually selected and further analyzed by PCR and sequencing to determine the genotype.

Candidate clones were grown to approximately 95% confluency in a 12-well plate (for adherent cells). Cells were then split into two new 12-well plates for each candidate clone, one for DNA isolation and the other for continued incubation. Cells were harvested for each clone, and the IndelCheck™ kit was used to screen for clones that contained an indel. The IndelCheck™ kit instructions were followed for DNA isolation or cell lysis, PCR amplification, and sequencing the product of the PCR.

### 4.5. High Throughput Screening

The 293T EGFP^KI^-LC3B cells were seeded at a level of 20,000 per well in a 96-well cell culture plate. On the second day, different concentration gradients of chloroquine (CQ), rapamycin, and DMSO were added to each well, and three replicate wells were set up for each concentration, in which chloroquine and rapamycin were used as positive control and DMSO was used as negative control. After 24 h, the cell culture medium was removed, and 100 μL of 4% paraformaldehyde was added to each well at room temperature for 10 min. Paraformaldehyde was then removed, and DAPI solution (1:3000) was added for 5 min at room temperature. The DAPI solution was removed, and cells were washed thoroughly (3×, 10 min each). After the sample was prepared, the 96-well plate was placed on a high-connotation instrument for photo analysis. The number of positive particles in each cell and the average fluorescence intensity of each cell were calculated. The number of cells per well was adjusted according to the analysis. The experiment was repeated three times to ensure the stability and repeatability of the experiment.

Based upon the results of the pre-experiment, in addition to the compounds in the drug library, positive and negative compounds should be added to each 96-well cell culture plate to ensure the accuracy and consistency of the experiment.

### 4.6. Immunofluorescence

HeLa cells were treated with HDYL-GQQ-495 for 24 h. Cells were then fixed with 4% paraformaldehyde in PBS buffer and incubated for 15 min at room temperature. Samples were blocked by incubation in 5% normal horse serum in PBS. P62 protein was detected using a P62 antibody (1:200, Proteintech) and the secondary antibody Brilliant Violet 421™ AffiniPure Donkey Anti-Rabbit IgG (H+L) (1:2000, Jackson ImmunoResearch). Samples were mounted in polyvinyl-based mounting media (Southern Biotech, Birmingham, USA) and imaged using an A1R Multiphoton laser confocal microscope (Nikon, Tokyo, Japan).

### 4.7. Microscale Thermophoresis Assay 

Binding affinities of HDYL-GQQ-495 with purified recombinant proteins MBP and MBP-P62 were measured by using the Monolith NT.115 (Nanotemper Technologies, Beijing, China). Proteins were fluorescently labelled according to the protocol of the RED-Tris-NTA (Nanotemper, no. MO-L1018) protein labelling kit. The labelled protein concentration used for each assay was 50 nM and mixed with the indicated concentrations of HDYL-GQQ-495 in the reaction buffer (20 mM HEPES, pH 8.0, 100 mM NaCl, 1 mM DTT, and 0.05% [*v/v*] Tween 20). The samples were loaded into premium capillaries (NanoTemper Technologies) after incubation at room temperature for 30 min and measured by using 40% MST power. Data analyses were performed using the Nanotemper analysis software (v.1.5.41) provided by the manufacturer, and the K_d_ was determined.

### 4.8. Synthesis HDYL-GQQ-495

Sodium hydride (0.6 g, 1.5 equiv.) suspended in 10 mL dry THF under N_2_ atmosphere. The 1,8-dihydroxyanthracene-9,10-dione **1** (2.4 g, 10 mmol, 1.0 equiv.) was dissolved in another 10 mL THF and was then added into the suspension of NaH slowly at 0 °C. After the addition was complete, the suspension was moved to room temperature, and stirring was continued for 0.5 h. Then dimethyl sulfate (2.4 mL, 2.5 equiv.) was added and refluxed overnight, and the reaction mixture turned into a yellow suspension. The reaction was stopped, quenched with sodium hydroxide aqueous solution, and then extracted with CH_2_Cl_2_. The solvent was reduced to afford 2.5 g crude product **2**.

Under an ice bath, 1,8-dimethoxyanthracene-9,10-dione **2** (2.5 g, 9.3 mmol) was dissolved in 25 mL concentrated sulfuric acid to obtain a dark purple solution, and the mixture of concentrated sulfuric acid and concentrated nitric acid (2.5 mL, V_H2SO4_:V_HNO3_ = 4:1) was added dropwise into the reaction mixture. Stirring was continued at 0 °C for 5 h until **2** was consumed, then the dark orange solution was poured in ice water with vigorous stirring. Large amounts of yellow sediments were precipitated and filtered, and the filter was washed with NaHCO_3_ aqueous solution for several times. The crude product was collected and dissolved in CH_2_Cl_2_ and was then purified by silica gel chromatography (PE: EA = 1:1) to obtain an orange solid **3** (2.3 g, 73% in two steps).

The 4,5-dimethoxy-1-nitroanthracene-9,10-dione **3** [41] (2.3 g, 7.3 mmol) was dissolved in the mixture of ethanol and ethyl acetate, 0.1 mL glacial acetic acid and 15% Pd/C (0.23 g, 10 wt%) were added, and then the reaction was stirred under a hydrogen atmosphere for 2 h. The reaction was stopped and filtered with the aid of silica gel, the filtrate was collected, and the solvent was reduced to afford a purple solid **4** (2.0 g, 94%). For ^1^H NMR (500 MHz, DMSO-d_6_), δ = 7.72–7.68 (m, 4H), 7.42–7.37 (m, 2H), and 7.13 (d, J = 9.4 Hz, 1H). For ^13^C NMR (125 MHz, DMSO-d_6_), δ = 184.3, 182.7, 158.7, 150.0, 147.1, 136.1, 134.3, 126.3, 123.2, 123.0, 122.7, 117.7, 117.4, 111.4, 57.8, and 56.6. For ESI-HRMS, [M + H]^+^ was calculated. For C_16_H_14_NO_4_^+^, *m/z* was 284.0917, and the value of 284.0921 was found.

The 1-amino-4,5-dimethoxyanthracene-9,10-dione **4** (28 mg, 1.0 mmol, 1.0 equiv.), 4-ethoxy-1-(4-(2-(piperidin-1-yl)ethoxy)phenyl)butane-1,3-dione **5** [41,42,43] (133 mg, 4.0 equiv.), and sodium acetate (29 mg, 3.5 equiv.) were put into a 10 mL one-neck round flask and heated to 125 °C for 12 h until the mixture turned yellow. The mixture solution of petroleum ether and ethyl acetate (V_PE_:V_EtOAc_ = 3:1) was added, and it was filtered to afford a yellow solid. Then the filter was collected and purified by silica gel chromatography (CH_2_Cl_2_: MeOH = 15:1 with 3% TEA) to gain crude compound **6**.

Compound **6** was dissolved in 1 mL glacial acetic acid, then 33% HBr in glacial acetic acid (100 equiv.) was added, and it was refluxed for 4 h. After the reaction was complete, it was cooled down to room temperature, and the solvent was reduced. The sediment was suspended in methanol and filtered to obtain a yellow solid, then washed with CH_2_Cl_2_ to afford the compound HDYL-GQQ-495 (36 mg, 70%). For ^1^H NMR (500 MHz, DMSO-d_6_), δ = 12.84 (s, 1H), 12.68 (s, 1H), 12.66 (s, 1H), 7.95 (d, J = 8.7 Hz, 2H), 7.82 (d, J = 9.1 Hz, 1H), 7.54 (t, J = 8.2 Hz, 1H), 7.43–7.47 (m, 2H), 7.10–7.14 (m, 3H), 4.49–4.45 (m, 2H), 3.56–3.48 (m, 4H), 3.03 (dd, J = 21.1, 9.5 Hz, 2H), and 1.88–1.32 (m, 6H). For ^13^C NMR (125 MHz, DMSO-d_6_), δ = 195.0, 190.3, 163.2, 162.3, 159.7, 136.6, 133.0, 132.9, 131.3, 129.4, 122.9, 120.2, 120.1, 115.3, 109.8, 62.6, 54.7, 52.7, 22.4, and 21.1. For ESI-HRMS, [M+H]^+^ was calculated. For C_30_H_27_N_2_O_6_^+^, *m/z* was 511.1864, and the value of 511.1865 was found.

### 4.9. Statistical Analysis

The expression levels of target genes were represented as normalizations to the levels of GAPDH using the 2^−∆∆CT^ method. Statistical analysis was performed using the GraphPad Prism 8.0 software (San Diego, CA, USA). The fluorescence intensity and immunoblot band staining were measured using ImageJ. A two-tailed Student’s *t* test was used to evaluate the group-level differences. Data were shown as the mean ± SEM. *** *p* < 0.001, ** *p* < 0.01, and * *p* < 0.05.

## Figures and Tables

**Figure 1 marinedrugs-21-00068-f001:**
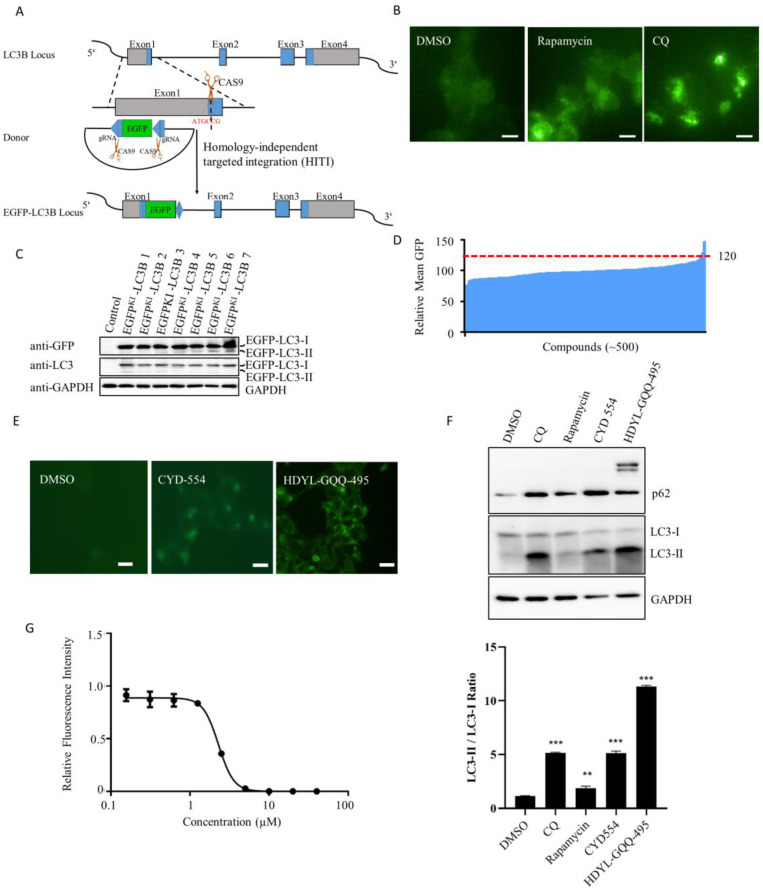
EGFP-LC3B knock-in cell lines were used to screen the library of natural products. (**A**) Schematic representation of the strategy for EGFP knock-in at the LC3 locus via the CRISPR/Cas9 system and HITI. Coding and non-coding regions of LC3 exons are shown with blue and gray bars, respectively. The EGFP sequence is shown in green. Scissors indicate the cutting site of Cas9. The pUC19-EGFP donor is the HITI template. (**B**) Representative images of EGFP puncta in the indicated cell lines with the indicated treatment. Scale bars, 10 μm. (**C**) Western blotting analysis of the expression of LC3 in the indicated 293T cells. Cell lysates were analyzed by Western blot using anti-LC3 or anti-GFP antibodies. (**D**) The relative mean GFP values higher than 120 were selected. (**E**) Representative pictures of indicated compounds with high GFP intensity in 293T EGFP^KI^-LC3B cells. Scale bars, 10 μm. (**F**) Western blotting analysis of the expression of LC3 and P62 in the HeLa cells. Cell lysates were analyzed by WB with anti-LC3, P62, and GAPDH antibodies (upper panel). Statistical analysis of the LC3-II/LC3-I ratio in HeLa cells with the indicated treatment. Data are shown as the mean ± SEM (*n* = 3). ** *p* < 0.01, *** *p* < 0.001, compared with DMSO (lower panel). (**G**) IC_50_ of HDYL-GQQ-495 on 293T EGFP^KI^-LC3B cells, IC_50_ = 2.30 ± 0.17 µM.

**Figure 2 marinedrugs-21-00068-f002:**
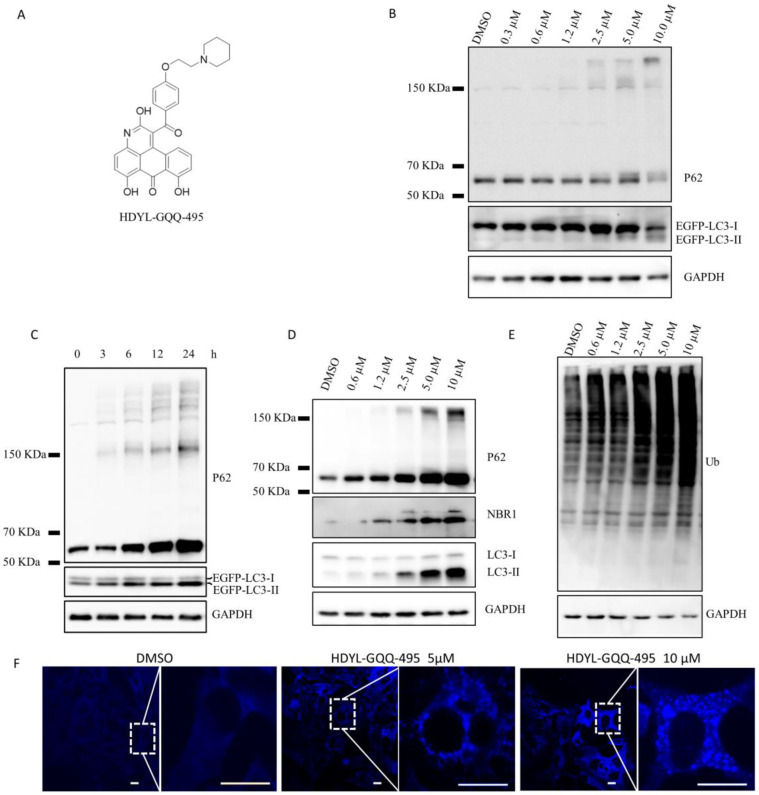
HDYL-GQQ-495 is an autophagy inhibitor. (**A**) Chemical structure of HDYL-GQQ495. (**B**) The 293T EGFP^KI^-LC3B 5 cells were treated with HDYL-GQQ-495 at the indicated concentration for 24 h and immunoblotted with anti-LC3, P62, and β-Tubulin antibodies. GAPDH was immunoblotted as a loading control. (**C**) The 293T EGFP^KI^-LC3B 5 cells were treated with 10 μM HDYL-GQQ-495 for the indicated time and immunoblotted with anti-LC3, P62, and GAPDH antibodies. GAPDH was immunoblotted as a loading control. (**D**) HeLa cells were treated with HDYL-GQQ-495 with the indicated concentration for 24 h. Cell lysates were analyzed by WB with anti-LC3, P62, NBR1, and GAPDH antibodies. GAPDH was immunoblotted as a loading control. (**E**) The 293T EGFP^KI^-LC3B 5 cells were treated with HDYL-GQQ-495 at the indicated concentration for 24 h and immunoblotted with anti-ubiquitin, GAPDH, and β-Tubulin antibodies. GAPDH and β-Tubulin were immunoblotted as loading controls. (**F**) HeLa cells were treated with the indicated concentration for 24 h. Cells were fixed and stained with antibodies for P62. Scale bars, 5 μm.

**Figure 3 marinedrugs-21-00068-f003:**
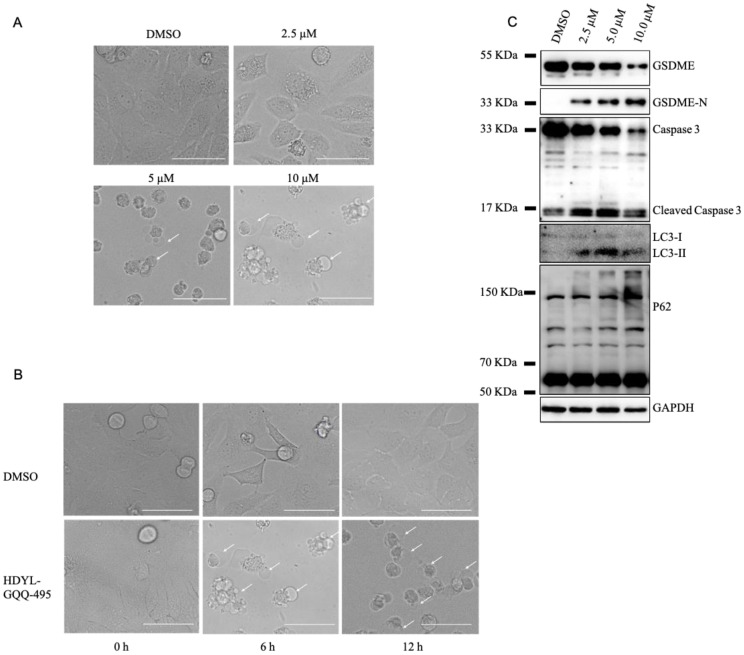
HDYL-GQQ-495 promotes pyroptosis. (**A**) Representative images of HDYL-GQQ-495 with the indicated concentration for 12 h. Scale bar, 100 μm. (**B**) Representative images of HeLa cells treated with 5 µM HDYL-GQQ-495 for the indicated time. Scale bar, 100 μm. (**C**) HeLa cells were treated with HDYL-GQQ-495 with the indicated concentration for 24 h. Cell lysates were analyzed by WB with anti-LC3, P62, GSDME, Caspase-3, and GAPDH antibodies. GAPDH was immunoblotted as a loading control.

**Figure 4 marinedrugs-21-00068-f004:**
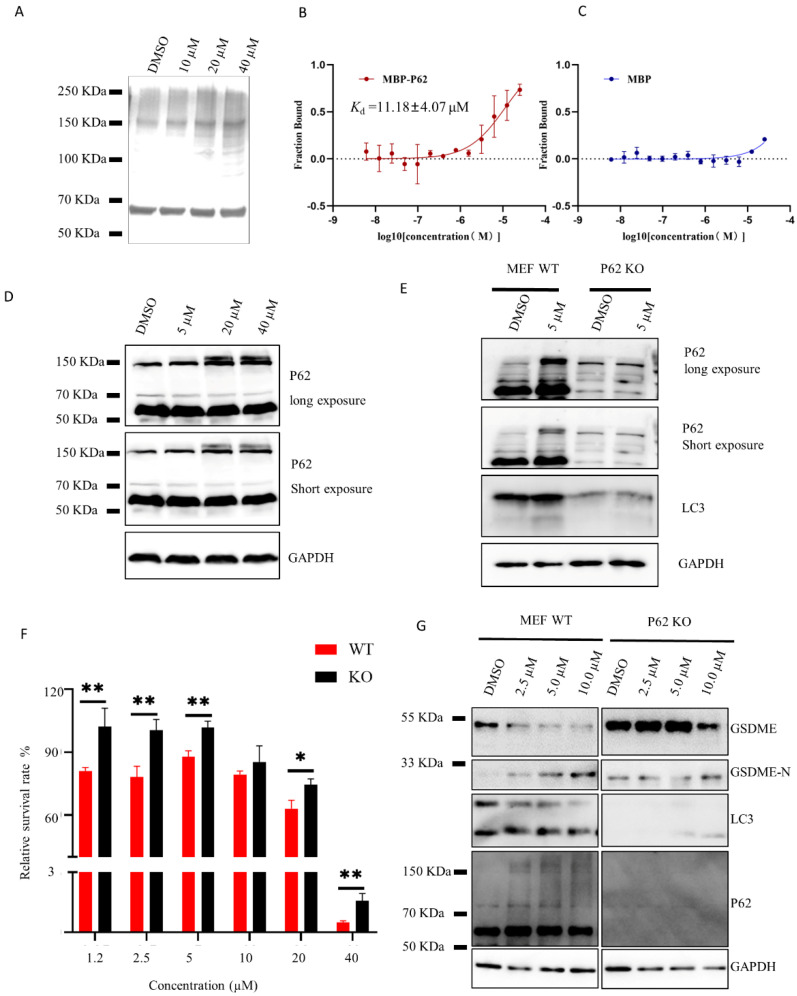
HDYL-GQQ-495 inhibits autophagy by inducing P62 oligomerization. (**A**) Purified P62 protein were treated with HDYL-GQQ-495 at the indicated concentration overnight, then purified P62 protein was added to 6x loading buffer (NO SDS and NO β-Me), and samples were subjected to 10% SDS-PAGE and stained with Coomassie blue. (**B**,**C**) Binding of HDYL-GQQ-495 to MBP or MBP-P62 in standard treated capillaries measured by microscale thermophoresis in which the concentration of labeled protein was kept constant. The binding curves and K_d_ values for MBP-P62 and HDYL-GQQ-495 calculated by the Nanotemper analysis software (v.1.5.41) are indicated. The K_d_ values of MBP and HDYL-GQQ-495 could not be determined. (**D**) The 293T cell lysate or lysate buffer were treated with HDYL-GQQ-495 at the indicated concentration overnight, then cell lysates were analyzed by Western blot with anti-P62 and GAPDH antibodies. GAPDH was immunoblotted as a loading control. (**E**) MEF WT or P62 KO cells were treated with 5 μM GQQ-495 for 24 h. Cell lysates were analyzed by Western blot with anti-P62, LC3, and GAPDH antibodies. GAPDH was immunoblotted as a loading control. (**F**) Relative survival of HDYL-GQQ-495 on MEF WT or P62 KO cells for 12 h. Data are shown as the mean ± SEM (*n* = 3). * *p* < 0.05, ** *p* < 0.01. (**G**) MEF WT and P62 knockout cells were treated with HDYL-GQQ-495 with the indicated concentration for 12 h. Cell lysates were analyzed by WB with anti-LC3, P62, GSDME, Caspase-3, and GAPDH antibodies. GAPDH was immunoblotted as a loading control.

**Table 1 marinedrugs-21-00068-t001:** Primers for sgRNA, donor, and genotyping target sequence.

Primer	Sequence
LC3B knock-in sgRNA	ACGTTCGGCTAAGATGCCGTCGG
Puc19-hindIII-EGFP-LC3-Foward	GACCATGATTACGCCAAGCTTCCGACGGCATGGTGCAGGGATCTGTGAG-CAAGGGCGAGGAGC
Puc19-EcoRI-GFP-LC3-Reverse	AAAACGACGGCCAGTGAATTCAGATCCCTGCACCATGCCGTCGGTCTTGTACAGCTCGTCCATGCC
Genome sequencing Forward1	CTATCGCCAGAGTCGGATTCGC
Genome sequencing Reverse1	TCACGGCGTCCCAGGCCCTG
Genome sequencing Forward2	TGGCTATCGCCAGAGTCGGA
Genome sequencing Reverse2	CTCCTCTCGACCGAGGCACT

## Data Availability

The data presented in this study are available on request from the corresponding author.

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
