# Peer review of "Marine-Derived Natural Product HDYL-GQQ-495 Targets P62 to Inhibit Autophagy"

_marinedrugs, 2023, doi:10.3390/md21020068_

Round 1

Reviewer 1 Report

In this work, the authors used a CRISPR/CAS9 knock-in strategy to construct endogenous autophagy reporter cell lines and uncovered a marine-derived compound targeting P62 polymerization to inhibit autophagy and promote pyroptosis. This work is meaningful and I think it can be published. Meanwhile, some issues need to be improved before publication in the following:

Reference 8 lacks article information.

References 25,37,47,48 also lack article information

The overall research work focuses on biological experiments, and chemical information is too little to be supplemented. Marine drugs also need innovation in chemistry. They screened a compound library of approximately 500 marine natural products and analogues to investigate molecules that alter the EGFP fluorescent. They identified 8 potential candidates which enhanced EGFP fluorescence and HDYL-GQQ-495 was the leading one.  The information of 500 molecules and 8 candidate molecules should be given. It is better to provide screening process and activity data of compounds, if too much can be put in SI.   

The synthesis process, melting point and NMR data of compounds 2-4, 6 and HDYL-GQQ-495 need to be provided.

Reviewer 2 Report

Comments to AUTHOR:

Your articles provide relevant information about the Marine-derived natural product to inhibit autophagy. However, there are some issues that need to be modified in the results and discussion section. So, I think this article should be revised by following suggestions.

Specific recommendations

Abstract

Line 27: significantly increased p-value?

Results

In cases when there is a significant change, P-values should be mentioned. For instance, lines 153 in chapter 2.1, line 183, 189, and 194 in chapter 2.2, and other chapters in the results section.

Line 135: DAPI staining should be added to determine the distribution of green fluorescence in the cytoplasm.

Lines 136-138: Add conclusion source images and statistical analysis of fluorescence intensity should be performed.

Line 153: Which one is LC3-Ⅱ and which one is LC3-Ⅰ?

Lines 195-196: Use multicell plots and then multicell plots for local zoom so that it is representative.

Lines 213-216: This should be in the introduction section.

Lines 221-227: This should be in the introduction section.

Lines 243-245: Results should be described in more detail.

Discussion

The discussion section must be restructured. Rather than describing the results again, this section draws your conclusion and then compares them with the previously published scientific literature results to support your conclusions.

Lines 281-296: The first paragraph of the discussion section must describe the answer to your hypothesis.

Lines 297-308: This part appears to be more of an introduction or background section.

Lines 328-330: The results must be compared with previous studies to support your results.

Materials and Methods

Lines 409-419: You should show the table with the primer sequences of the experiment.

Lines 470-471: Please determine the calculation of relative gene expression, whether 2-△△CT or 2-CT, and I didn't see the experimental content about the genes.
